# The Role of Technology in Online Health Communities: A Study of Information-Seeking Behavior

**DOI:** 10.3390/healthcare12030336

**Published:** 2024-01-29

**Authors:** LeAnn Boyce, Ahasan Harun, Gayle Prybutok, Victor R. Prybutok

**Affiliations:** 1Department of Advanced Data Analytics, Toulouse Graduate School, University of North Texas, Denton, TX 76201, USA; 2Department of Information Systems, Robert C. Vackar College of Business, University of Texas Rio Grande Valley, Edinburg, TX 78539, USA; ahasan.harun@utrgv.edu; 3Department of Rehabilitation and Health Services, College of Health and Public Service, University of North Texas, Denton, TX 76203, USA; gayle.prybutok@unt.edu; 4Department of Information Technology and Decision Sciences, G. Brint Ryan College of Business, University of North Texas, Denton, TX 76203, USA; victor.prybutok@unt.edu

**Keywords:** online forum, information-seeking behavior, online information seeking, online information-seeking behavior, online health information, online health communities

## Abstract

This study significantly contributes to both theory and practice by providing valuable insights into the role and value of healthcare in the context of online health communities. This study highlights the increasing dependence of patients and their families on online sources for health information and the potential of technology to support individuals with health information needs. This study develops a theoretical framework by analyzing data from a cross-sectional survey using partial least squares structural equation modeling and multi-group and importance–performance map analysis. The findings of this study identify the most beneficial technology-related issues, like ease of site navigation and interaction with other online members, which have important implications for the development and management of online health communities. Healthcare professionals can also use this information to disseminate relevant information to those with chronic illnesses effectively. This study recommends proactive engagement between forum admins and participants to improve technology use and interaction, highlighting the benefits of guidelines for effective technology use to enhance users’ information-seeking processes. Overall, this study’s significant contribution lies in its identification of factors that aid online health community participants in the information-seeking process, providing valuable information to professionals on using technology to disseminate information relevant to chronic illnesses like COPD.

## 1. Introduction

The role of technology in healthcare continues to evolve, with online health communities (OHCs) emerging as a powerful platform for sharing knowledge and promoting collective action [1]. In particular, Facebook groups have become a significant source of health-related information, providing a sense of community and belonging for individuals facing medical challenges. However, several issues persist with OHCs, and there is a need for greater understanding and management of these communities. This study offers valuable insights into the factors that trigger contributors’ online information-seeking behaviors within OHCs, specifically in the context of COPD, a chronic and incurable respiratory condition with significant economic and societal impact. The findings of this study bridge a critical gap in the understanding of OHCs and underscore the crucial role of technology in facilitating access to information and support for those in need, ultimately improving outcomes and reducing costs. This study’s contribution is particularly significant in the context of Information Technology and People, focusing on technology, as it highlights the potential of technology to support those with significant medical challenges through innovative approaches to disseminating relevant information.

This study presents two primary objectives that significantly deviate from those of previous studies. First, this research assists healthcare professionals in enhancing their approach to COPD OHCs by considering age and gender factors. The proposed modifications are expected to optimize available resources and improve patient outcomes. Second, this study employs an importance–performance map analysis (IPMA) at the construct and indicator levels to obtain valuable insights into critical concerns related to online health information-seeking behavior. By comparing participants’ comprehensive experiences with the average scores derived from latent variables that detail performance, the IPMA evaluates the importance of participants’ involvement in the endogenous construct [2,3].

The following research questions (RQ) address these objectives and the research gap:RQ1:Do age and gender influence online health information-seeking behavior?RQ2:Do age and gender relate to external factors such as self-worth, perceived experience, perceived usefulness, and perceived ease of use?

This research sheds light on the utilization of current technologies by the public to fulfill their health information needs. It fills a significant research gap by enhancing the understanding of how medical professionals can serve their patients better by gaining insight into how their patients utilize technology. The findings of this study are expected to have significant implications for health practitioners, policymakers, and researchers alike, emphasizing the importance of incorporating age and gender factors in the design and deployment of online health information resources.

## 2. Literature Review

The Internet has become an essential resource for individuals seeking health information, with millions of people globally relying on online sources for guidance. The Digital 2022 Global Overview Report [4] indicates that around thirty-six percent of Internet users are actively searching for health information, and Foster’s report [5] confirms this high engagement with health content on social media. On Facebook, with 2.91 billion users, over 1.8 billion engage in health-related groups monthly, forming over 10 million communities [6]. Jia, Pang, and Liu [7] found that over a quarter of health information consumers search for information online multiple times daily. Facebook groups, defined as communities offering belonging and connection, became critical support networks during the COVID pandemic, with most users participating in mutual support [8].

The relevance of online health communities is on the rise, but challenges remain. This study contributes to the literature on online health behavior triggers and the influence of disease-specific factors, with a focus on chronic illnesses like COPD. Our findings bridge a critical gap in the understanding of online health communities and offer actionable information for both medical and non-medical professionals. Moreover, this study highlights how technology aids in information dissemination and support network formation. The economic implications of COPD, costing USD 49.0 billion in 2020, are also addressed [9]. Despite OHCs’ extensive use for various health issues (e.g., mental health [10,11], AIDS/HIV [12,13], and cancer [14,15]), their role in COPD management has been underexplored. These platforms offer not just disease-specific information but also emotional support and social interaction (refs. [16,17,18,19]), which are key to patient empowerment and improved quality of life. This study underscores technology’s potential to reduce COPD’s financial and societal impacts by connecting patients and facilitating information access.

### 2.1. Theoretical Background and Hypotheses

Our research team developed items based on a five-point Likert scale drawn from the extant literature to assess the survey constructs. In Table 1, the sources for the survey items are provided. 

For the relationships between the exogenous factors and the outcome variable (Figure 1), we propose the following hypotheses:

**H1.** 
*Perceived ease of use (PEOU) plays a significant positive role in shaping the information-seeking behaviors of COPD forum users.*


**H2.** 
*Perceived usefulness has a significant positive effect on information-seeking behaviors.*


**H3.** 
*Perceived expertise (PE) plays a significant positive role in shaping the information-seeking behaviors of COPD forum users.*


**H4.** 
*Sense of self-worth (SSW) plays a significant positive role in shaping the information-seeking behaviors of COPD forum users.*


**H5.** 
*In the context of information-seeking behavior, the influence of determinants (PU, PEOU, SSW, and PE) is moderated by gender.*


**H6.** 
*In the context of information-seeking behavior, the influence of determinants (PU, PEOU, SSW, and PE) are moderated by age.*


### 2.2. Roles of Perceived Ease of Use and Perceived Usefulness (H1 and H2)

In this study, information-seeking effectiveness is defined as the comprehensive assistance provided to those with specific medical conditions, encompassing social and emotional support and critical health information based on the accuracy of information, details concerning rehabilitation, and access to other pertinent services [24]. Recent work has shown how online health information impacts patient decision-making and proactive engagement in health management within online health communities (OHC) [25]. The Technology Acceptance Model (TAM), developed by Davis [26], assesses how individuals perceive and adopt new technologies by examining perceived usefulness and ease of use [27]. OHCs facilitate patient and healthcare provider interaction via the Internet [28]. Research based on TAM suggests that perceived ease of use and usefulness positively influence technology use [29]. Perceived usefulness is defined as the extent to which an individual believes that utilizing a specific system or technology, such as an Online Health Community (OHC), will enhance the overall quality of their life. For a system, including an OHC, to be effective, it must either enhance or assist its users, consequently influencing the extent to which users actively contribute within the designated online platform, such as a Facebook group. On the other hand, perceived ease of use refers to the degree to which an individual perceives the online system as effortless to operate [30]. A good technological infrastructure, a favorable attitude toward technology, and a user-friendly, uncomplicated interface are anticipated to enhance the likelihood of adopting and using online resources. TAM’s insights are instrumental in understanding interactions within OHCs, especially regarding technology’s perceived benefits [30].

### 2.3. Role of Perceived Expertise (H3)

Perceived expertise in online health forums is the belief in one’s capability to positively influence health outcomes. It is a key predictor of participation in online health communities, with a proven link between cancer management program involvement and perceived expertise [31]. Those with higher expertise are more likely to engage in their own disease management, utilizing various resources. The Internet’s role as a primary health information source has been extensively researched. Lee, Niederdeppe, and Freres [32] note that the wealth of information online helps fill knowledge gaps, reducing feelings of uncertainty and despondency, particularly concerning COPD. The availability of such information has been linked to greater patient control, satisfaction, and empowerment and enhanced physician communication [33]. Consequently, this study examines the correlation between participants’ perceived expertise in COPD Facebook groups and their online information-seeking behavior. 

### 2.4. Role of Sense of Self-Worth (H4)

In this study, self-worth is operationalized as individuals’ perception of their value addition to an online community by sharing knowledge [34]. This is based on Social Exchange Theory (SET), which explains social behavior as a series of transactions where participants engage in the reciprocal exchange of goods, which can manifest as either non-material or material entities [35], thus establishing equilibrium between the rewards gained and the costs incurred within these interactions. Yan et al. [21] view knowledge sharing as an exchange where the costs and benefits can be balanced. Here, an information need is any query requiring a response, with OHCs providing patients with the opportunity to obtain timely and effective answers to their questions, especially when access to their physicians is limited [36]. Social support is characterized by positive conversations that contribute to the well-being of participants, facilitating interactions among members who share similar illnesses. The multifaceted nature of social support includes attributes such as companionship, emotional support, and opportunities for socialization [37]. Sense of self-worth within SET is an individual’s perceived impact on the group through their knowledge contributions [34,38]. Costs in OHCs, within the framework of SET, include the cognitive efforts of recalling past experiences, such as emotions like irritation, pain, and depression, and executional resources like time, money, and materials [39]. Additionally, participants’ engagement in online communities is reinforced by their perceived elevated status within groups, which, in turn, boosts overall participation [38]. 

### 2.5. Role of Gender (H5)

Prior research on the moderating effect of gender on behavioral intention within diverse online environments has produced inconsistent results across different and, at times, similar applications. Researchers like Lian and Yen [40] and Tan and Ooi [41] report that gender does not moderate associations of perceived ease of use and perceived usefulness with users’ behavioral intentions. Similarly, Kim [42] and Wong et al. [43] were not able to establish the moderating effect of gender on consumers’ use of hotel email and tablet apps. Conversely, Mandari and Chong [44] and Acheampong et al. [45] report that the association between behavioral intention (BI) and PU was greater in male users, but the association between BI and ease of use was not as strong for males about mobile payments and mobile government service usage. Such findings, in the research of Tarhini et al. [46], were partly confirmed in the case of online learning. Their research discovered that associations between the adoption of eLearning technologies by students and perceived usefulness were unvarying between females and males. However, the association among eLearning adoption intention and perceived ease of use was greater for females [46]. The moderating effect of gender on online technologies and the use of information is therefore somewhat subjective and necessitates more examination in the context of COPD forum participants. 

### 2.6. Role of Age (H6)

Age-related variances among humans within technology use are influenced by self-efficacy and life experience. Research indicates that older adults may feel too old to learn new technological skills, unlike younger adults, who are more eager to engage with and learn from new technologies [47,48]. However, the age-related effects on online behavior are not uniform. Tarhini et al. [46] found that in the context of eLearning, perceived usefulness correlates more with younger users’ adoption intentions, while ease of use is more significant for older users. Similarly, Liebana-Cabanillas et al. [49] observed that while expertise, usefulness, and trust have less of an influence on older adults’ purchase intentions online, ease of use affects both age groups equally. Lian and Yen [40] reported that age does not have any moderating impact on the usefulness of online shopping or on perceived ease of use (PEOU). In contrast, Kim [42] and Tan and Ooi [41] found no age moderation in the adoption of hotel tablet apps and online shopping or hotel tablet app adoption. Therefore, the moderating impact of age is relatively subjective and necessitates further attention in the context of COPD forum users. 

## 3. Materials and Methods

### 3.1. Setting and Participants

To identify relevant online COPD communities, we conducted a systematic search for the keyword “COPD” in Facebook groups in August 2020. Our search yielded 95 groups that were specifically related to COPD. These groups varied in their objectives, ranging from providing emotional support to disseminating information about pulmonary rehabilitation, exercise, diet, and treatment options. Some groups also aimed to raise awareness of the disease and advocate for improved patient care. The membership of these groups ranged from 16 to 13,000 individuals, and the number of posts per day varied from 0 to 60.

### 3.2. Data Collection

Before collecting the data, the survey was reviewed by survey research experts and members of a PhD student research team with expertise in survey methods. This questionnaire was developed to collect data to measure the relationships in the proposed model to analyze the results between COPD Facebook groups. A 5-point Likert scale (5 = strongly agree, 4 = somewhat agree, 3 = neither agree nor disagree, 2 = somewhat disagree, and 1 = strongly disagree) was selected to measure the responses for all constructs. 

The survey was revised based on the constructive feedback these individuals provided. IRB approval was obtained from the university, and a pilot test was conducted, resulting in slight re-framing and adjustments in the survey questions to improve the general clarity of the questionnaire. There were originally fifty questions, and after the review and pilot run, eleven of the questions were deleted. The survey was entered into Qualtrics. Please see the Appendix A for the questions included in the survey. 

To gather survey data, a licensed respiratory therapist on our research team accessed the identified Facebook COPD groups and posted the survey link with the group’s approval. Out of the 66 groups contacted, 46 granted permission to post the survey link, and we successfully posted it in 32 groups. This approach allowed us to collect valuable data from online COPD OHCs and gain insights into the research questions that we aimed to address.

### 3.3. Analysis

Several analyses were conducted in this study on information-seeking behavior in COPD OHCs. These include:

Sample Size Determination: The determination of the sample size for the model was based on OLS regression properties. 

Common Method Bias: Kock’s conservative method was implemented to ensure that variance inflation factor values were below the threshold of five.

Non-Response Bias: A post hoc test comparing early and late respondents was carried out using an independent samples t-test.

Partial Least Squares Structural Equation Modeling (PLS-SEM): This was used for data analysis to develop a predictive model of information-seeking behavior, due to its appropriateness for predictive studies, stability with smaller sample sizes, and efficiency in analyzing models with convergence issues and complicated structural relationships.

Reflective Measurement Model: The reliability and validity of the constructs were evaluated using Dillon–Goldstein’s rho for internal consistency and average variance extracted (AVE) for construct validity. 

Structural Model Evaluation: The model’s predictive value was assessed using a variance inflation factor (VIF), bootstrapping methods, and blindfolding for cross-validation.

Multi-Group Analysis: Measurement invariance across gender and age groups was tested using the Measurement Invariance of Composite Models (MICOM) process. The analysis allowed a comparison of path coefficients across groups since partial measurement invariance was achieved. Importance–performance map analysis (IPMA) was implemented to assess the diagnostic value of the model, focusing on ‘information-seeking behavior’ and its associations with other exogenous constructs. It also evaluated the importance and performance of these constructs within the structural model.

These analyses were integral to the study’s aim of understanding and predicting information-seeking behavior within online COPD communities, and they provided a comprehensive evaluation of the model’s reliability, validity, and predictive power.

## 4. Results

Determination of the sample for the model was based on the OLS regression properties [50]. Thus, this research required forty-one observations for identifying values of approximately 0.25, at the five percent significance level, with a statistical power of eighty percent [51]. After cleaning the data, there were two hundred and one usable responses for analysis. Thus, with a sample of two hundred and one, the minimum sample size to represent the population was far exceeded. The demographics of the participants were as follows: The proportion of females was seventy-eight percent and of males was twenty-two percent. Eleven percent of the study participants were between the ages of thirty-one to fifty-four years, forty-six percent were between the ages of fifty-five and sixty-four, thirty-five percent were between the ages of sixty-five and seventy-four, and eight percent were aged seventy-five years and above. The majority of the participants had less than fifty thousand dollars of yearly income.

Common method bias was addressed because participants’ anonymity was assured, and all responses were de-identified before the data analysis. Common method bias was addressed by utilizing Kock’s conservative method [52]. Kock proposed allowing a variance inflation factor under the threshold of five, which is also achieved in this study. A post hoc test was conducted to determine whether non-response bias could affect the generalizability of our findings. To accomplish this, early and late respondents were compared. As established in the research of Li and Calantone [53], the first seventy-five percent of the survey participants were designated early respondents, and the last twenty-five percent were designated late participants. No significant difference was found after comparing early and late respondents using an independent samples t-test. Thus, non-response bias was dismissed.

The data were analyzed using partial least squares structural equation modeling (PLS-SEM) for the following reasons. The primary objective of this study is to develop a predictive model of information-seeking behavior within COPD online communities. PLS-SEM is particularly appropriate for circumstances where prediction is the goal. Secondly, partial least squares structural equation modeling (PLS-SEM) employs a system of ordinary least squares regression that remains stable with smaller sample sizes, a finding confirmed through simulation analysis by Reinartz et al. [54]. Thirdly, relative to a covariance-based analysis, PLS-SEM is efficient in analyzing models where convergence may be a problem [50,55]. Wold’s [56] research emphasized that PLS-SEM is optimal for a complicated structural relationship. Lastly, PLS path modeling integrates the values of latent variables, which are required for importance–performance map analysis. 

### 4.1. Reflective Measurement Model

The reliability evaluation of each construct in the measurement model is also shown in Table 2. For assessing the reliability of internal consistency, the Dillon–Goldstein method was used, which additionally considers outer loadings as additional indicators. All constructs have composite reliability values over 0.7, and internal consistency reliability is also confirmed. 

An analysis of the average variance extracted (AVE) and the indicator reliability was also conducted to determine the validity of the reflective measurement model. Following the deletion of the items that failed to achieve the recommended value of 0.7, Table 2 presents the relevant themes of the retained survey items. While it is not recommended, an AVE value equal to or greater than 0.5 is considered acceptable because it suggests that the construct can explain over half of the variance associated with the indicator. Table 2 demonstrates that all constructs meet the 0.5 minimum value for AVE. Therefore, we assume that the convergent validity of our survey instrument is acceptable.

By analyzing indicator cross-loadings, we determined that discriminant validity was upheld. However, cross-loadings cannot indicate a lack of discriminant validity if two constructs are completely correlated. In addition, Table 3 also demonstrates that the indicators are accurate according to the Fornell–Larcker criterion [57]. The upper portion of Table 3 shows that the square root of the AVEs, shown on the diagonals for each construct, is greater than the correlations between the other latent variables. However, the Fornell–Larcker criterion performs poorly if construct indicator loadings differ slightly. 

As a result, the heterotrait–monotrait correlation coefficients (HTMT) procedure [58] was applied and all the relationships were under the accepted value of 0.90, thus reinforcing discriminant validity (see the bottom portion of Table 3). Distribution was tested using bootstrapping to ensure that it was consistent with HTMT statistics. The confidence interval obtained from 5000 bootstrap samples substantiates that the HTMT values are significantly different, reinforcing discriminant validity. Therefore, the constructs are empirically distinct.

### 4.2. Structural Model Evaluation

The variance inflation factor (VIF) evaluated each construct for collinearity. Collinearity among the constructs is eliminated since the VIF values are below the threshold of five. Based on the bootstrap percentile confidence intervals, we determined whether the model’s results were statistically significant (bias-corrected). Following Preacher and Hayes [59], 5000 bootstrap samples were run, with the original sample number of observations being included in each bootstrap sample. Table 4 illustrates the structural model relationships based on the 5000 bootstrap samples. Sixty-point two percent of the variation within the endogenous construct-information-seeking behavior is explained by the model. 

Next, to assess the cross-validated redundancy, a blindfolding procedure with a distance of six as our predetermined distance was implemented. In other words, the combination of the in-sample and out-of-sample predictive powers should be higher than zero for an endogenous construct to define the predictive accuracy of a structural model [55]. In this research, the calculated statistic produced a value greater than zero. Thus, it was concluded that the model has predictive value. Additionally, when comparing the statistical values of information-seeking behavior with Hair’s recommendations [55], it is apparent that the in-sample predictive power for information-seeking behavior was higher than the moderate level. 

To assess the out-of-sample predictive power of the model, PLS predictive analysis was conducted with the default settings of ten folds and ten repetitions [60]. To analyze the results, the mean absolute prediction error (MAPE) values from both the PLS and LM analyses were examined, as well as the root mean square error (RMSE) and the predicted values from the PLS analysis. As can be seen in the lower portion of Table 3, all the values in the PLS analysis were greater than zero, indicating that the prediction errors created by the PLS-SEM results were less than the prediction errors created solely by relying on mean values. Additionally, the out-of-sample predictive power level was high regarding RMSE values at the indicator level since all three items of information-seeking behavior in the PLS-SEM model resulted in fewer prediction errors than the LM benchmark. At the indicator level, two of the three items exhibited similar behavior, indicating an acceptable degree of predictive power.

The above results confirm the influences of the constructs perceived ease of use, perceived usefulness, perceived expertise, and sense of self-worth on information-seeking behavior (H1, H2, H3, and H4).

### 4.3. Multi-Group Analysis

Invariance was evaluated using the Measurement Invariance of Composite Models (MICOM) process, which has three stages: 1. configural invariance evaluation, 2. compositional invariance evaluation, and 3. evaluation of the similarity of mean and variance values. In the event that stage 3 is not satisfied, a multi-group analysis can still be conducted [50]. In this case, partial measurement invariance is attained [50]. The PLS path models of this study, data treatments, and group-specific model approximations were the same about the algorithmic situations utilized for both gender and age groups (<65 years and 65 years+). Thus, configural invariance is confirmed. Compositional invariance was evaluated utilizing 1000 permutations [61] at the five percent significance level. The findings indicated that the *p* values were higher than 0.05, and the correlation was not substantially lower than 1, which validates compositional invariance. In the evaluation of the uniformity of variance and means throughout all age groups, we found that the permutation *p* values for the means in all constructs were higher than 0.05 and also higher for the variances for the information seeking, perceived expertise, and sense of self-worth constructs. For gender, the *p*-values for variances for all constructs were greater than 0.05, but all composite means were lower than 0.05. 

Although full measurement invariance was not determined, the path coefficients for both gender and age groups can be compared since partial measurement invariance was determined at stages 1 and 2 [50]. To comply with the stringent guidelines for power analysis [50] and identify an R squared value of 0.25 at the one percent significance level and an eighty percent power level, forty-one observations per group were required. Adhering to these guidelines, the group sample sizes of one hundred and fifty-seven females and forty-four males are adequate. These guidelines are also met for age groups in that our study includes one hundred and fifteen participants who are sixty-four years old or younger and eighty-six participants who are sixty-five years old or older. 

Table 5 shows the differences between male and female users in the following cases:

Perceived expertise has a stronger effect on information-seeking behavior for female participants than males. The impact of perceived expertise on the information-seeking behavior of male participants is insignificant.

Sense of self-worth has a stronger influence on information-seeking behavior for male participants than females, as there is no influence for female participants.

These results partly confirm that the influences of perceived expertise and sense of self-worth on information-seeking behavior are moderated by gender, partially supporting H5, since this is not the case with the effects of the constructs of perceived ease of use and perceived usefulness on information-seeking behavior.

In addition, Table 5 shows the following: Perceived expertise has a strong effect on information-seeking behavior for people who are sixty-four years old or younger. The impact of perceived expertise on information-seeking behavior in people who are sixty-five years old or older is statistically insignificant. The same was found for the relationship between a sense of self-worth and information-seeking behavior.

Perceived usefulness has a strong effect on information-seeking behavior for people who are sixty-five years old or older but not for those who are sixty-four years old or younger.

Thus, age moderates the influences of perceived expertise, perceived usefulness, and sense of self-worth on information-seeking behavior. Age does not moderate the relationship between information-seeking behavior and perceived ease of use. Therefore, H6 is partially supported.

### 4.4. Importance–Performance Map Analysis (IPMA)

To assess the diagnostic value of our models, a post hoc study employing the IPMA was conducted as proposed by Martilla and James [62]. The evaluation was based on the PLS estimates, emphasizing the importance of each construct in the existing relationships, and average values denoting performance. Specifically, the IPMA focused on the final main construct, ‘information-seeking behavior,’ examining its associations with other exogenous constructs and the performances of the currently hypothesized relationships within these exogenous experiences.

The total effects of predominant relationships within the structural model were evaluated and revealed the variance of the main construct, information-seeking behavior. Before calculating the averages of each indicator to represent performance, dissimilar scores of each of the latent variables and the indicators with scores between 0 to 100 were standardized [63]. Figure 2 demonstrates that, at the construct level, perceived expertise is located on the far left of the graph. This means that this construct is of lesser significance regarding information-seeking behavior relative to the other constructs. Figure 2 shows that perceived ease of use is situated on the far-right section of the graph. This indicates that information seekers in the Facebook COPD online community deem perceived ease of use as the most significant factor. 

## 5. Discussion

As we transition to discussing the implications of our findings, it is essential to highlight the pivotal role of interface design in user engagement within disease-specific online health communities (OHCs). Figure 3 emerges as a crucial element in our analysis, illustrating the areas within forum design that require enhancement to facilitate improved information-seeking behavior. This section will delve into the nuances of these findings, exploring the impact of perceived ease of use, usefulness, and self-worth on user participation. Moreover, we will examine how demographic variables such as age and gender differentially shape information-seeking activities, underscoring the importance of tailored approaches in the development and management of OHCs. Our discussion will draw upon these insights to propose practical strategies for medical professionals and forum administrators to foster a supportive and effective environment for patients and caregivers in these digital spaces.

Although many studies have been conducted on various online health communities, little research has focused on creating and managing disease-specific online health communities. As a result, more information is needed on the factors that trigger online information-seeking by participants within OHCs. This research investigated the extent and manner in which exogenous factors within a disease-specific OHC influence participants’ online information-seeking behaviors.

Relevant constructs from the existing literature were utilized to develop and evaluate this theoretical framework. Consequently, this study sheds new light on the information-seeking behavior of a disease-specific community. Thus, this research provides a valuable perspective to medical professionals on implementing the proposed outline, which can increase the quality of life of patients, their caregivers, and their families. 

According to this analysis, perceived ease of use is the strongest predictor of information-seeking behavior. Consequently, perceived ease of use should be given high priority. Additionally, perceived usefulness and sense of self-worth correlate positively and have significant predictive power. Further, our results confirm a positive relationship between perceived expertise and information-seeking behavior. These results provide insight into the thought processes of the forum participants and emphasize the need to focus on the participants’ experiences to achieve a successful outcome. A systematic evaluation of the comparative effects of exogenous factors on information-seeking behavior within disease-specific Facebook groups is provided by this research.

This research provides a theoretical framework emphasizing the importance of concentrating on these factors to design a successful approach to positively impacting disease-specific OHCs. To facilitate decision-making, the constructs should be considered within the context of an integrated model, such as the one developed in our study. Based on IPMA, this research provides insight into how to encourage disease-specific Facebook group participants’ information-seeking behaviors. As shown in Figure 2, it is pertinent to emphasize that perceived expertise is less significant than other factors when considering information-seeking behaviors. In addition, the sense of self-worth construct performs significantly below average, as indicated on the IPMA’s *y*-axis. For this reason, concentrated effort is necessary to improve its performance. In the future, additional research should be conducted to improve the construct’s present performance for perceived ease of use in light of its placement on the *y*-axis.

Information seeking is influenced most by the indicator which corresponds to skillfully searching COPD-related information (PEOU3). As a result, it is imperative to continue focusing on improving its current performance. In addition, there is a need to enhance the opportunities for interaction between participants (PEOU1). This is crucial since this indicator is the second most significant factor. Currently, however, it is only performing at an average level, as illustrated by the *y*-axis in Figure 3. In the same way, Figure 3 shows how forums (PEOU2) can be improved significantly by improving navigation. As a result, it is critical to pay attention to those factors that contribute significantly to the information-seeking behavior of forum participants.

Further insights are provided by the secondary analysis from a gender-based perspective. Males and females have substantially different associations between perceived expertise and information-seeking behavior. In addition, the effect of sense of self-worth significantly differs between males and females. Information-seeking behaviors are influenced differently by perceptions such as perceived usefulness and perceived expertise between participants who are 64 years old or younger and participants who are 65 years old or older. Medical and non-medical professionals can use these insights to provide a pleasant experience for disease-specific Facebook participants while searching for information. While a specific feature may be prioritized to improve performance, the time and money needed to encourage such actions may not be worthwhile. Thus, it is possible to develop a comprehensive plan for practical approaches by using this blueprint: 

Implement the model as a benchmark in various online forums by distributing survey questions to provide a benchmarking environment.Develop a plan for forum administrators and moderators, which encourages frequent exchanges with participants and positively impacts the thought processes of disease-specific information seekers.

Additional textual data were collected that allowed the completion of a qualitative analysis for comparison with the findings. The qualitative analysis supported the main themes identified in the current research. For example, it was found that gender differences existed, and females tended to seek information more than males. In addition, revisiting the site using this mixed method approach allowed us to obtain longitudinal data and confirmed the stability of the findings over time. So, the finding that females reach out more than males was again confirmed.

## 6. Limitations and Future Work

This study presents a theoretical evaluation framework for information-seeking behavior in disease-specific online forums and evaluates the impact of age and gender on user behavior within the context of the model. Despite the limitations of self-reported data collected via online questionnaires [64], this study significantly contributes to the understanding of disease-specific support groups on Facebook. Future research should analyze other disease-specific online health communities both within and outside of Facebook using this model. Additionally, implementing other advanced statistical models, such as multilevel modeling (MLM) or latent growth modeling, would help us to gain a greater understanding of the complexities of information-seeking behaviors within OHCs. Lastly, consideration of the education level, socioeconomic status, or health literacy of those surveyed would provide a broader range of moderating variables.

## 7. Conclusions

The study employed cross-sectional survey data analyzed using partial least squares structural equation modeling, multi-group analysis, and importance–performance maps, resulting in the validation of the proposed model. The statistical methods used in this study ensured that the predictions made by the research are reliable. The research concluded with significant findings, notably that age and gender influence online health information-seeking behavior. It was discovered that perceived expertise and sense of self-worth differed based on the way each gender seeks information. Specifically, perceived expertise is more influential for women, while sense of self-worth is more influential for men. Lastly, this study highlights that while age moderates how perceived expertise, usefulness, and self-worth influence this behavior, it does not affect the impact of perceived ease of use.

The knowledge gained from this study is crucial for creating and managing online health communities (OHCs), with implications for both medical professionals and non-medical professionals. Medical professionals can recommend credible online health communities to patients and provide them with an “information prescription” for optimal patient outcomes. Technology professionals can use this study’s findings to develop novel approaches for disseminating relevant information to individuals with chronic diseases, such as COPD.

Moreover, the study highlights the potential of technology in improving outcomes for caregivers, patients, and their families. Forum administrators and moderators can use our findings to enhance the interaction opportunities, navigation, and perceived expertise of community members, thereby positively impacting information-seeking behaviors.

In conclusion, this study provides significant insights into the information-seeking behavior of disease-specific forum users, with implications for the development and management of OHCs. Our findings have important implications for both medical and technology professionals, highlighting the potential of technology to improve outcomes for individuals with chronic diseases.

## Figures and Tables

**Figure 1 healthcare-12-00336-f001:**
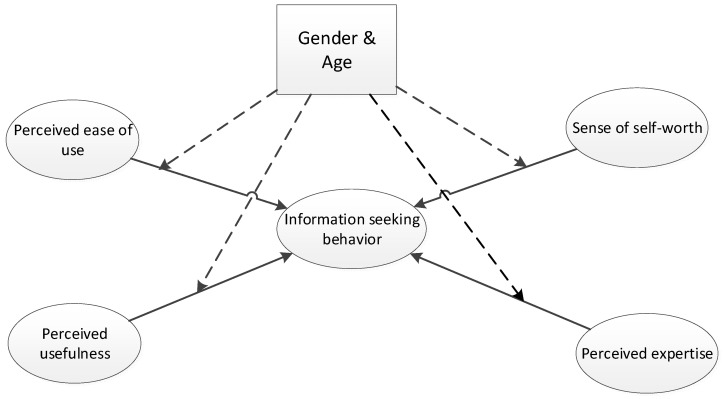
Theoretical framework.

**Figure 2 healthcare-12-00336-f002:**
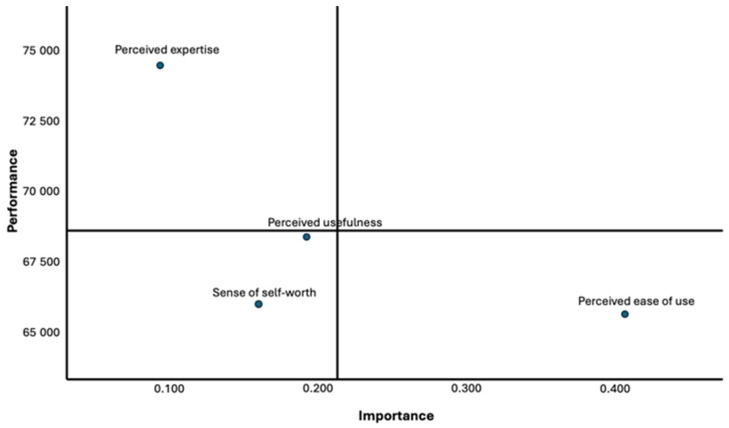
IPMA construct level.

**Figure 3 healthcare-12-00336-f003:**
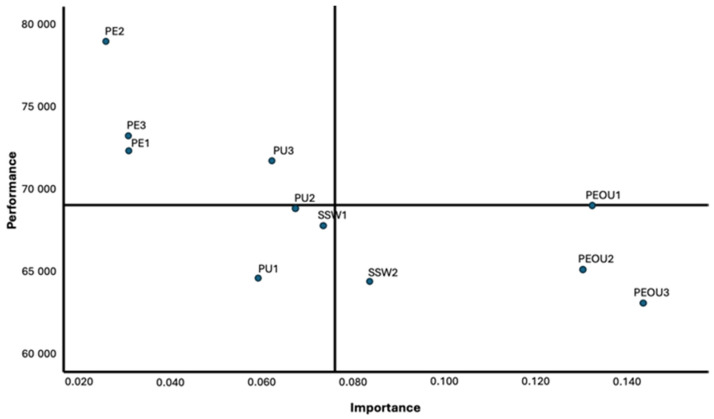
IPMA indicator level.

**Table 1 healthcare-12-00336-t001:** Survey constructs, composite reliability, and AVE scores.

Constructs	Item Sources	Item Label	Loadings	Dillon–Goldstein’s *p*	Average Variance Extracted (AVE)
Perceived ease of use	Ahadzadeh et al. [20]	PEOU1	0.775	0.889	0.728
		PEOU2	0.881		
		PEOU3	0.897		
Perceived usefulness	Ahadzadeh et al. [20]	PU1	0.864	0.917	0.786
		PU2	0.900		
		PU3	0.895		
Sense of self-worth	Yan et al. [21]	SSW1	0.868	0.863	0.759
		SSW2	0.874		
Perceived expertise	Durcikova et al. [22], Kollmann et al. [23]	PE1	0.801	0.880	0.709
		PE2	0.828		
		PE3	0.895		
Information-seeking behavior	Nambisan [16]	ISE1	0.881	0.900	0.750
		ISE2	0.880		
		ISE3	0.836		

**Table 2 healthcare-12-00336-t002:** Composite reliability and AVE scores.

Constructs	Dillon–Goldstein’s *ρ*	Average Variance Extracted (AVE)
Information-seeking behavior	0.900	0.750
Perceived ease of use	0.889	0.728
Perceived expertise	0.880	0.709
Perceived usefulness	0.917	0.786
Sense of self-worth	0.863	0.759

**Table 3 healthcare-12-00336-t003:** Discriminant validity (Fornell–Larcker and HTMT criteria).

**Fornell–Larcker Criterion**					
Constructs	Information-seeking behavior	Perceived ease of use	Perceived expertise	Perceived Usefulness	Sense of self-worth
Information-seeking behavior	0.866				
Perceived ease of use	0.719	0.853			
Perceived expertise	0.472	0.449	0.842		
Perceived Usefulness	0.635	0.622	0.460	0.886	
Sense of self-worth	0.561	0.549	0.423	0.505	0.871
**HTMT Criterion**					
Information-seeking behavior	~				
Perceived ease of use	0.876				
Perceived expertise	0.580	0.561			
Perceived Usefulness	0.745	0.748	0.553		
Sense of self-worth	0.741	0.738	0.571	0.657	~

~ indicates it is not possible to have HTMT value with itself.

**Table 4 healthcare-12-00336-t004:** Structural model results and out-of-sample predictive performance at indicator level.

Paths				Path Coefficient		Bias-Corrected 95% Confidence Interval	
Perceived ease of use → Information-seeking behavior				0.442 ***		[0.312, 0.578]	
Perceived expertise → Information-seeking behavior				0.099 **		[0.005, 0.192]	
Perceived usefulness → Information-seeking behavior				0.235 ***		[0.113, 0.363]	
Sense of self-worth → Information-seeking behavior				0.158 **		[0.031, 0.294]	
Items	PLS			LM			
	RMSE	MAPE	Q2_predict	RMSE	MAPE	RMSE_PLS_—RMSE_LM_	MAPE_PLS_—MAPE_LM_
ISE1	0.570	13.316	0.492	0.582	13.282	−0.012	0.034
ISE2	0.622	14.305	0.422	0.642	15.172	−0.020	−0.867
ISE3	0.623	14.512	0.388	0.644	14.614	−0.021	−0.102

*** *p* < 0.01; ** *p* < 0.05.

**Table 5 healthcare-12-00336-t005:** Path coefficients for gender and age.

**Paths for Gender**	**Path Coefficients (Male)**	**Bias-Corrected 95% Confidence Interval**	**Path Coefficients (Female)**	**Bias-Corrected 95% Confidence Interval**
Perceived ease of use → Information-seeking behavior	0.332 ***	[0.093, 0.560]	0.487 ***	[0.345, 0.635]
Perceived expertise → Information-seeking behavior	0.080	[−0.104, 0.233]	0.117 *	[0.001, 0.229]
Perceived usefulness → Information-seeking behavior	0.353 ***	[0.093, 0.591]	0.192 ***	[0.060, 0.335]
Sense of self-worth → Information-seeking behavior	0.289 *	[0.012, 0.598]	0.089	[−0.059, 0.230]
**Paths for Age**	**Path Coefficients (64 Years or Less)**	**Bias-Corrected 95% C.I.**	**Path Coefficients (65+ Years)**	**Bias-Corrected 95% C.I.**
Perceived ease of use → Information-seeking behavior	0.468 ***	[0.304, 0.648]	0.420 ***	[0.225, 0.598]
Perceived expertise → Information-seeking behavior	0.162 **	[0.022, 0.284]	0.013	[−0.126, 0.145]
Perceived usefulness → Information-seeking behavior	0.085	[−0.060, 0.240]	0.408 ***	[0.218, 0.580]
Sense of self-worth → Information-seeking behavior	0.220 **	[0.032, 0.413]	0.091	[−0.051, 0.261]

*** *p* < 0.01; ** *p* < 0.05; * *p* < 0.10.

## Data Availability

Dataset available on request from the authors.

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
