# Peer review of "The Role of Technology in Online Health Communities: A Study of Information-Seeking Behavior"

_healthcare, 2024, doi:10.3390/healthcare12030336_

Round 1
Reviewer 1 Report
Comments and Suggestions for Authors
Good effort.
Please consider on the following issues:
1. Method: This part is very short. Please write in detail about your method. How did you analyze your data? Describe your survey. Please use a flow diagram to describe the method.
2. Discussion: Please mention other related papers. Please mention the novelty of your study and its strengths.
3. Please check reference numbering (double numbering).
4. Conclusion: Please summarize your conclusion according to your objective and results.
Author Response
Thank you for the valuable feedback. I am attaching a document with our point-by-point responses.
Kindly,
LeAnn Boyce

Reviewer 2 Report
Comments and Suggestions for Authors
I am grateful for the opportunity to review this manuscript on study of information-seeking behavior. Please find my suggestions and questions.
MANUSCRIPT
The text is very long, making it difficult for readers. It should be better organized and summarized.
ABSTRACT
Lines 12-13: “This research article significantly contributes to both theory and practice by providing valuable insights into the role and value of healthcare in the context of online health communities. Is this manuscript a research article?
INTRODUCTION
Lines 58-60: RQ1: How does age and gender influence online health information-seeking behavior? RQ2: How do age and gender relate to external factors such as self-worth, perceived experience, perceived usefulness, and perceived ease of use?
In the research questions, “how” is more used for qualitative studies (interviews, focus groups...). Please consider this.
Literature review
-The text is long and tiring to read. The authors could rewrite it.
-Please always write the text the same way:
For example, Line 82: “… more than twenty-five percent of consumers … and line 88: “… 91% of respondents reported ...”
-Lines 136-141: “Before collecting the data, the survey was reviewed by survey research experts and members of a PhD student research team with expertise in survey methods. The survey was revised based on the constructive feedback these individuals provided. IRB approval was obtained from the university, and a pilot test was conducted, resulting in slight reframing and adjustments in survey questions to improve the general clarity of the questionnaire.” This information could be written in Materials and Methods section.
-Lines 158-303: This text could be more condensed. Part of the text could be used to compare the results of this work with the literature.
MATERIALS AND METHODS
An online survey was used. How many questions were asked? What kind of answers were available? The authors could describe the questionnaire and make it available in the Appendix section.
RESULTS
-Line 327: “Eleven percent of the study were between the ages..”. This could be: …Eleven percent of the study PARTICIPANTS were between the ages…
-Lines 329-330: “…between the ages of sixty-five and seventy-four, and eight percent were seventy-four years and above.” seventy-four???? This could be: … between the ages of 65-74, and 8% were 75 years and above…
-Please always write the text the same way:
For example: Line 327: “of the participants were females 78% and males 22%.” … and line 328: “…thirty-one to fifty-four years, forty-six percent…”
- Page 11. Table 3: Please look at the table configuration. Is it ok?
- Lines 440-453: The first information in Table 4 is focused on gender. However, the authors first described the age in the text. // Lines: 454-464: The second information in Table 4 focuses on age. However, the authors described the gender second in the text. In this way, the authors presented results on H6 before the results on H5. Please rewrite this.
- Lines 451-453: “Age does not moderate the relationship between information-seeking behavior and perceived ease of use. Therefore, H6 is partially supported.” …. 461-463: “These results partly confirm that the influences of perceived expertise and sense of elf-worth on information-seeking behavior are moderated by gender, partially supporting H5 since this is not the case… What about H1, H2, H3 and H4?
DISCUSSION
The discussion section needs to be rewritten:
-Please add a summary of results in the first paragraph.
-Line 524. “Figure 3 illustrates how forums (PEOU2) can be improved significantly by improving…” Where is Figure3?
REFERENCES
-We can observe the duplication of articles. Examples: 8 and 9, 32 and 33, 45 and 46, 58 and 59, 77 and 78, 80 and 81, 83 and 84… Please correct this.
-The authors could add references from the year 2023.
Please be careful to change reference numbers in the text.
Author Response

(The authors gave the same response as above.)

Reviewer 3 Report
Comments and Suggestions for Authors
1. In the Introduction section, specifically within the first paragraph, the author discusses the evolving role of technology in healthcare and the emergence of online health communities (OHCs) as influential platforms for knowledge sharing and collective action. Providing appropriate references to support these statements and acknowledging the sources contributing to this understanding would be beneficial.
2. In section 2.1, "Theoretical Background and Hypotheses," where you mentioned, "The research team developed items based on a 5-point Likert scale," it would be beneficial to provide a proper reference for this statement.
3. I recommend you consider employing advanced statistical methods, such as latent growth modeling (LGM) or multilevel modeling (MLM), to better understand information-seeking behavior's complex and dynamic aspects within online health communities. By employing these advanced statistical methods, you can uncover nuanced patterns of information-seeking behavior, explore how these behaviors change over time, and better understand the factors influencing them. These approaches offer a more comprehensive and sophisticated analysis of online health community dynamics, ultimately enhancing the depth of your research findings.
4. To ensure the stability and reliability of the proposed model, I recommend conducting additional model validation techniques, such as cross-validation or bootstrapping. These techniques can help assess the model's robustness by testing its performance on different subsets of the data or by generating multiple samples from the original data. By doing so, the study can ensure that the model does not overfit the data and that the findings are generalizable to the population of interest.
5. To gain a more comprehensive understanding of the factors influencing information-seeking behavior, I recommend exploring potential moderating variables beyond gender and age. This could involve investigating variables such as education level, socioeconomic status, or health literacy, which may significantly shape individuals' information-seeking behaviors within online health communities. By considering a broader range of moderating variables, the study can capture a more nuanced and detailed picture of the complex dynamics in this context.
6. While the authors have already identified the use of partial least squares structural equation modeling (PLS-SEM) and Importance-performance map analysis, I suggest employing a mixed-methods approach. Combining quantitative data with qualitative insights can provide a more comprehensive understanding of information-seeking behavior in online health communities.
By incorporating qualitative methods, such as interviews, focus groups, or content analysis of forum discussions, your study can capture rich contextual details and personal experiences that quantitative data alone may not fully elucidate. This approach allows for a deeper exploration of the motivations, challenges, and perceptions of individuals engaging in information-seeking behaviors within online health communities.
This holistic research approach enriches the understanding of this complex phenomenon, complementing the quantitative findings with qualitative narratives. It provides a well-rounded view of the subject matter, shedding light on the "how" and "why" behind the observed behaviors and contributing to a more nuanced and insightful analysis.
However, the authors can add my suggestion in the future research section.
7. In the discussion section, it would be beneficial for the authors to conduct a comparative analysis of their research findings in relation to relevant studies in the field. Specifically, there are some related papers:
1. Zhu, P., Shen, J., & Xu, M. (2020). Study on the evolution of information sharing strategy for users of online patient community. Personal and Ubiquitous Computing, 1-7.
2. Basoglu, N., Daim, T. U., Atesok, H. C., & Pamuk, M. (2010). Exploring the impact of information technology on health information-seeking behaviour. International Journal of Business Information Systems, 5(3), 291-308.
3. Zhang, Y., Lee, E. W., & Teo, W. P. (2023). Health-Seeking Behavior and Its Associated Technology Use: Interview Study Among Community-Dwelling Older Adults. JMIR aging, 6, e43709.
4. Walsh, A. M., Hamilton, K., White, K. M., & Hyde, M. K. (2015). Use of online health information to manage children’s health care: a prospective study investigating parental decisions. BMC health services research, 15(1), 1-10.
5. Pálsdóttir, Á. (2009). Seeking information about health and lifestyle on the Internet. Information Research, 14(1), 2.
Author Response

(The authors gave the same response as above.)

Round 2
Reviewer 1 Report
Comments and Suggestions for Authors
The present version is fine. I encourage the authors to write the points (red font in the conclusion) as a paragraph (not 1. 2. 3..).
Author Response
Hello Reviewer 1,
Thank you for taking your valuable time to give feedback on our article. We have made the following changes you have suggested:
| Reviewer Comment | Change in Manuscript | Lines in Manuscript |
| The present version is fine. I encourage the authors to write the points (red font in the conclusion) as a paragraph (not 1. 2. 3..). |
The research concluded with significant findings, notably that age and gender influence online health information-seeking behavior. It was discovered that perceived expertise and sense of self-worth differed on the way genders seek information. Specifically, perceived expertise is more influential for women, while sense of self-worth is more influential for men. Lastly, this study highlights that while age moderates how perceived expertise, usefulness, and self-worth influence this behavior, it does not affect the impact of perceived ease of use. |
545 - 551 I have highlighted this revision in the manuscript |
Thank you for this opportunity to improve the manuscript.
Kindly,
LeAnn Boyce
Reviewer 3 Report
Comments and Suggestions for Authors
The authors have diligently addressed and revised all of the reviewer's concerns, demonstrating a thorough and meticulous approach to enhancing the quality of their research. I appreciate the conscientious point-by-point revisions, which have significantly strengthened the manuscript. I have no objection to the revised version, and based on the comprehensive improvements made, I believe the manuscript is now well-suited for publication in the Healthcare.
Author Response
Hello Reviewer 3,
Thank you for your words of encouragement. We appreciate your feedback and agree your comments made the paper stronger.
Kindly,
LeAnn Boyce